# Insecticidal Effect of Wild-Grown *Mentha pulegium* and *Rosmarinus officinalis* Essential Oils and Their Main Monoterpenes against *Culex pipiens* (Diptera: Culicidae)

**DOI:** 10.3390/plants11091193

**Published:** 2022-04-28

**Authors:** Amal Ramzi, Abdelhakim El Ouali Lalami, Yassine Ez zoubi, Amine Assouguem, Rafa Almeer, Agnieszka Najda, Riaz Ullah, Sezai Ercisli, Abdellah Farah

**Affiliations:** 1Laboratory of Applied Organic Chemistry, Faculty of Sciences and Technologies, Sidi Mohamed Ben Abdellah University, Imouzzer Street, Fez 2202, Morocco; ramzi.amal@usmba.ac.ma (A.R.); eloualilalami@yahoo.fr (A.E.O.L.); y.ezzoubi@uae.ac.ma (Y.E.z.); farah.abdellah1@gmail.com (A.F.); 2Higher Institute of Nursing Professions and Health Techniques, Regional Health Directorate, EL Ghassani Hospital, Fez 30000, Morocco; 3Biotechnology, Environmental Technology and Valorization of Bio-Resources Team, Department of Biology, Faculty of Science and Techniques Al-Hoceima, Abdelmalek Essaadi University, Tetouan 2117, Morocco; 4Laboratory of Functional Ecology and Environment, Faculty of Sciences and Technologies, Sidi Mohamed Ben Abdellah University, Imouzzer Street, Fez 2202, Morocco; 5Department of Zoology, College of Science, King Saud University 2455, Riyadh 11451, Saudi Arabia; ralmeer@ksu.edu.sa; 6Department of Vegetable and Herbal Crops, University of Life Sciences, Lublin 50A Doswiadczalna Street, 20-280 Lublin, Poland; agnieszka.najda@up.lublin.pl; 7Department of Pharmacognosy, College of Pharmacy, King Saud University, Riyadh 11451, Saudi Arabia; rullah@ksu.edu.sa; 8Derpartment of Horticulture, Faculty of Agriculture, Ataturk University, 25240 Erzurum, Turkey; sercisli@gmail.com

**Keywords:** mosquitoes, *Mentha pulegium*, *Rosmarinus officinalis*, essential oils, monoterpenes, fumigant toxicity, *Culex pipiens*

## Abstract

The present study investigates the insecticidal effect of plant extract such as *Mentha pulegium* and *Rosmarinus officinalis* essential oils and some of their major compounds; these plants are well known for their many biological activities. The fumigant toxicity was evaluated, using glass jars, against female adults of *Culex pipiens* that constitute a mosquito vector of important diseases such as the West Nile virus. The adulticidal test showed that both essential oils and monoterpenes presented an insecticidal effect better than the chemical insecticide (Deltamethrin). The highest mortality percentages for the two essential oils have occurred at 312.5 µL/L air (between 56.14 ± 1.7% and 97.71 ± 3.03% after 24 h and 48 h of treatment). Moreover, all tested monoterpenes (carvone, R(+)-pulegone, 1,8-cineole, camphor and α-pinene) have produced high mortalities that varied depending on the time of the treatment and the concentrations used. Lethal concentrations (LC_50_) obtained for the essential oils and the main compounds have also varied according to the exposure time. *M. pulegium* and *R. officinalis* essential oil exhibited the lowest LC_50_ values after 24 h (72.94 and 222.82 µL/L air, respectively) and after 48 h (25.43 and 55.79 µL/L air, respectively) while the pure molecules revealed the lowest LC_50_ values after 48 h (between 84.96 and 578.84 µL/L air). This finding proves that the two essential oils and their main compounds have an insecticidal potential, which could help to develop natural toxic fumigants that may be used as an eco-friendly alternative in integrated and sustainable vector management.

## 1. Introduction

Nowadays, plant-secondary metabolites (essential oils and extracts) have attracted scientific committees since the isolated compounds such as terpenoids, flavonoids, phenolics and alkaloids have exhibited many biological activities [1,2,3]. Essential oils constitute a great interest for researchers who always try, through their scientific studies, to discover the great potential of these substances as antioxidant, antimicrobial, anti-inflammatory and insecticidal agents. In fact, plants from the Lamiaceae family are well known for their biologically active essential oils, and many reports have documented the presence of various compounds such as terpenes, iridoids, flavonoids and phenolic constituents in species of this family [4]. Moreover, it has been reported that the two Lamiaceae plants, *Mentha pulegium* and *Rosmarinus officinalis* have many important biological activities [5]. *M. pulegium* possesses several biological properties; this aromatic plant has proven antioxidant and anticholinesterase [6], anthelmintic [7], antimicrobial [4] and insecticidal effects [8,9,10]. Similarly, *R. officinalis* is well known for its beneficial effect as a therapeutic agent [11] and it also displays insecticidal potential [12].

The insecticidal effect of these medicinal plants has been highly reported against stored product insects [13,14,15,16,17], and as larvicidal and repellent agents against some mosquitoes [18,19,20,21]. However, there were a few reports on the fumigant activity of these plants on mosquito adults, especially *Culex pipiens* that is considered a mosquito vector of important diseases and well known for their public health and veterinary importance [22]. In addition, few studies have reported the insecticidal effect of monoterpenes against adults of *C. pipiens* [23], yet the insecticidal effect of these compounds was reported against other insect genera [24,25].

Indeed, *Culex pipiens* play a major role in the transmission of harmful viruses that represent a serious threat to public and veterinary health [26]. This mosquito especially induces the West Nile fever disease that has occurred by a virus from the Flaviviridae family; it could affect the nervous system leading to severe symptoms that may induce death [27]. Hence, plant-based natural products could represent an alternative approach for sustainable control since the wide appearance of resistance of *C. pipiens* adults to synthetic insecticides [28,29].

Therefore, this paper aims to investigate the insecticidal potential of *M. pulegium* and *R. officinalis* essential oils against female adults of *C. pipiens*, and also to highlight the comparative toxicity of their main compounds.

## 2. Material and Methods

### 2.1. Material

To evaluate the insecticidal effect, *Mentha pulegium* and *Rosmarinus officinalis* were collected during their flowering period (between July and September for *M. pulegium* and between January and May for *R. officinalis*) from a mountainous zone in Taounate town (northeastern Morocco). The samples were identified at the National Agency for Medicinal and Aromatic Plants in Taounate, Morocco. The pure compounds; 1,8-cineole (99%), camphor (98%), α-pinene (98%), carvone (98%) and R(+)-pulegone (99%) were purchased from Sigma-Aldrich (Steinheim, Germany). Deltamethrin (0.05%) was used as a positive control at the diagnostic dose suggested by the World Health Organization for mosquito adults.

### 2.2. Essential Oils Isolation and Chemical Analysis by GC-MS

Essential oils (EOs) were obtained from the aerial parts of each plant (stem, flowers and leaves) after their hydrodistillation for 3 h using a Clevenger apparatus. The yield of both EOs was calculated based on the dry weight of the plants. Gas chromatography coupled with mass spectrometry (GC-MS) was adopted to determine the aroma profile of each EO. To elucidate the process, a Hewlett-Packard (HP 6890) Gas Chromatographer coupled with a Mass Spectrometer (HP 5973) was used for the chemical analyses required. The Gas Chromatography was equipped with the column HP-5MS (30 m × 0.25 mm, film thickness 0.25 µm). Helium was chosen as carrier gas with a flow rate of 1.4 mL/min. The split mode was used, and the injection temperature was fixed from 50 to 200 °C at a heating rate of 4 °C/min, for 5min. The mass spectrometry apparatus was operated at specific conditions, which are the ionisation energy (70 eV); the ionisation source temperature (280 °C), the solvent cut time (3 min) and the mass spectra were recorded over a range of 30 to 1000 atomic mass units at 0.5 s/scan. The compounds were identified based on their retention indices, by comparison with the NIST MS Search database 2012, and by Adams terpene library [30].

### 2.3. Culex Pipiens Rearing Conditions

The collection of larvae was performed randomly using a rectangular plastic tray from a site characterized by a high larval density of *Culex pipiens* (*C. pipiens*), it is called Oued El Mehraz and it is situated in an urban area of Fez City (north-east of Morocco). To obtain mosquito adults for ulterior use in bioassays, collected larvae were kept under specific rearing conditions; an average temperature of 22.6 °C ± 2 °C, relative humidity of 70% ± 5%, with dimension cages (24 × 24 × 24 cm). Adults were nourished on sucrose solution. *C. pipiens* adults were morphologically identified using the Moroccan key of Culicidae identification [31], and the identification software of mosquitoes of Mediterranean Africa [32].

### 2.4. Fumigant Toxicity

The Fumigant activity was evaluated as described by Zahran and Abdelgaleil (2011) [23] with slight modifications; it was carried out on female adults of *C. pipiens* (2–3 days) using glass jars. A series of concentrations were prepared for each EO and they were expressed per volume of air (19.53, 39.06, 78.125, 156.25, 312.5 µL/L air) and also for the five pure compounds (50, 100, 156.25, 312.5, 625, 1250, 2500, 5000 µL/L air). Then, 5µL of each concentration was applied on Whatman filter papers that were attached to the lower surface of the jar covers. DMSO was used as a negative control, while Deltamethrin was used as a positive control. Three replicates (20 females per test) were performed. The mortality percentages of the adults exposed to the insecticidal substances were recorded after 24 h and 48 h of treatment; they were calculated using Abbott’s formula (1). Probit analysis was used to estimate LC_50_ values.
(1)$$\%\;Mortality\;Corrected=\left[\frac{\%\;Mortality\;Observed-\%\;Mortality\;Control}{100-\%\;Mortality\;Control}\right]\times 100$$

### 2.5. Statistical Analyses

Mortality rates data of both EOs and their main compounds were analyzed statistically by analysis of variance (ANOVA) using IBM SPSS 21 software. LC_50_ values were estimated according to Finney’s mathematical methods [33].

## 3. Results

### 3.1. Extraction Yield and Chemical Composition

The hydrodistillation of the two aromatic plants provided a variable amount of EOs since the yield of *M. pulegium* was 2.26 ± 0.14%, while it was 1.96 ± 0.33% for *R. officinalis*. Gas chromatography and mass spectrometry revealed that 18 molecules (93.35%) were identified in *M. pulegium* EO; it was dominated by pulegone (74.03%), carvone (5.45%) and dihydrocarvone (3.66%), whereas *R. officinalis* EO contains 23 constituents representing 94.23%. 1,8-cineole (29.31%), camphor (24.66%) and α-pinene (12.76%) were identified as the major components in this EO (Table 1).

### 3.2. Comparative Toxicities of Both EOs and the Pure Compounds on C. pipiens Adults

The fumigant toxicity results indicated that the mortality of *C. pipiens* adults differs between each EO tested and its main compounds, concentrations used, as well as the time of exposure; it could be said that the mortality was concentration and time-dependent. As illustrated in Figure 1, *M. pulegium* EO exerted a high insecticidal effect at all various concentrations tested than *R. officinalis* EO, yet they both have an important insecticidal potential towards adults of *C. pipiens* compared to the chemical insecticide (Deltamethrin). It appears that the mortality efficacy increases with the increase in concentrations and exposure time. At the highest concentration (312 µL/L air), *M. pulegium* EO killed 78.94 ± 0.16% and 97.71 ± 3.03% of mosquitoes after 24 h and 48 h of treatment, respectively, while the mortality rates recorded for *R. officinalis* were calculated at 56.14 ± 1.7% and 75.94 ± 1.92%, respectively. In contrast, the Deltamethrin displayed mortality of 19.29 ± 0.01% and 34.05 ± 1.01% after the same exposure periods.

Figure 2 shows the comparative toxicity of monoterpenes at different concentrations. Apparently, the toxicity varies from one compound to another and depends on the concentration applied. Nevertheless, they all have toxic effects on *C. pipiens* adults. At the lowest concentration (50 µL/L air), 1,8-cineole, camphor and α-pinene that constitute the main compounds in *R. officinalis* EO were found to be more effective than carvone and R(+)-pulegone; the principal constituents in *M. pulegium* EO, after 24 h and 48 h of exposure. It also seems that the five monoterpenes exhibited a high toxic effect as the concentrations increased. At 5000 µL/L air, all compounds produced high mortality that varies between 48.63 ± 1.3% and 61.5 ± 0.5% after 24 h and 68.24 ± 0.57% and 90.55 ± 1.2% after 48 h. 1,8-cineole, camphor and α-pinene were the most effective at this highest concentration after 48 h, while 1,8-cineole and camphor displayed high mortalities that reached 84.33 ± 1.54% and 90.55 ± 1.2%”. It could be concluded that these monoterpenes have the potential to kill *C. pipiens* adults even at lower concentrations.

Table 2 summarizes the lethal concentration values, and other parameters obtained from regression lines of the test of both essential oils and the five monoterpenes. It could be extrapolated that LC_50_ values for *M. pulegium* EO were estimated at 72.94 and 25.43 µL/L air after 24 and 48 h of treatment, respectively. Whereas their main monoterpenes (carvone and R(+)-pulegone), presented LC_50_ values of 1713.36 (1703.33–1723.36) and 5395.58 (5295.58–5494.68) µL/L air after 24 h and 538.96 (512.84–568.36) and 578.84 (578.24–597.64) µL/L air after 48 h, respectively. For *R. officinalis* EO, LC_50_ values were obtained at 222.82 (210, 234.29) and 55.79 (50.77–60.81) µL/L air after 24 and 48 h, respectively. However, 1,8-cineole, camphor and α-pinene displayed LC_50_ values at 5395.65 (5282.45–5486.55), 2269.64 (2190.41–2354.45) and 1294.64 (1230–1345.14) µL/L air after 24 h and 84.96 (75.8–95.26), 205.38 (167.83–300.51) and 85.74 (74.04–98.21) µL/L air after 48 h, respectively. From the results of mortality rates and LC_50_ values, it could be concluded that both EOs killed more female adults and revealed the lowest LC_50_ values in comparison to their main constituents.

## 4. Discussion

In mosquito management programs, control of mosquito adults is still hard given difficulties related to the accessibility to the rearing site, while the control of larvae is just limited to aquatic habitats [23]. Choosing the appropriate method and the insecticidal substances is of great priority in mosquito control. Therefore, this study has been achieved to valorize plant EOs and monoterpenes by evaluating their insecticidal effect against mosquito adults. Our findings indicated that *M. pulegium* and *R. officinalis* EOs exhibited toxic effects against *C. pipiens* female adults as well as their selected major compounds (1,8-cineole, camphor, α-pinene, carvone and R (+)-pulegone). Among the five pure molecules, 1,8-cineole, camphor and α-pinene that represent the main substances in *R. officinalis* EO were found to be more toxic, but generally, all compounds possess an insecticidal potential. In fact, all obtained results demonstrated that the two EOs presented fumigant toxicity better than their pure components. This variability in the insecticidal effectiveness could be explained by the fact that EOs are mixtures of chemical constituents and their biological effect could be attributed to the major monoterpenes components, or minor components, or probably the synergistic or antagonistic effect between various natural products, with no individual compound making a dominating contribution [35].

Recently, EOs are considered one of the best strategies in mosquito control, they have proven an effective mosquitocidal effect. The fumigant toxicity against adults of *C. pipiens* using the two EOs and even the five monoterpenes have not been documented in some countries including Morocco. However, the larvicidal activity of essential oils against *C. pipiens* has been reported [36,37]. Similarly, the larvicidal effect of monoterpenes such as 1,8-cineole, (R)-carvone and (R)-camphor was evaluated towards *C. pipiens* [23]. 1,8-cineole and camphene were also tested against larvae of *C. pipiens* pallens [38]. The larvicidal toxicity of pinenes (enantiomers of α- and β-) against the same mosquito species was also investigated by Michaelakis et al. (2009) [37]. Chen et al. (2021) [39] have also reported the insecticidal potential of fifteen pure compounds including (+)-camphor and 1,8-cineole on larvae of the beet armyworm *Spodoptera exigua* while they showed LC_50_ values of 161.22 (154.48–168.24) and 104.17 (98.71–109.94) μg/insect, respectively.

Worldwide, EOs isolated from Chinese plants revealed fumigant toxicity against *C. pipiens* adults such as *Mentha piperita*, which belongs to the same genus and was among the tested oils [40]. Additionally, EO obtained from an Egyptian plant which was *Mentha microphylla* a genus of the mint family (Lamiaceae), also has a proven fumigant effect against *C. pipiens* adults [41]. The fumigant effect of another genus, *Mentha longifolia*, has also been achieved against the females of *C. pipiens*. This plant with pulegone (74/95%), 1,8-cineole (7.35%) and eucarvone (2.68%) were the main compounds exerting toxicity of LC_50_ = 0.215 µL/L [42]. Our results are in agreement with those obtained previously by Zahran and his collaborators, in 2017 [43], who confirmed that *R. officinalis* was among the EOs that were the most potent fumigants against adults of *C. pipiens*. Furthermore, five essential oils including *Mentha piperita* were tested for their repellency effect against the adult females of *C. pipiens* [44].

The insecticidal effect of these EOs has also been reported against another insect genus; Brahmi and his coauthors (2016) [45] evaluated the contact toxicity, fumigant toxicity and repellency effect of *Mentha pulegium* and *Mentha rotundifolia* from Algeria against adults of *Rhyzopertha dominica*, the principal pest of wheat. Indeed, pulegone (70.4%) was the main component found in *M. pulegium* EO. The fumigant effect of both EOs revealed a mortality rate of 39.2% and 44.3%, respectively, at a concentration of 2000 µL/L. Similarly, Isikber et al., (2006) [46] evaluated the fumigant toxicity of *R. officinalis* EO against all life stages of *Tribolium confusum*, a common pest insect of stored flour and grain.

The fumigant toxicity of *M. pulegium* and *R. officinalis* EOs may be attributed to their major or minor monoterpenes as EOs contain different components that may act together on different ways. It had been reported that monoterpenes proved an insecticidal effect when they were tested alone, noting, for instance, certain main components such as α-pinene, limonene, α-terpineol, β-pinene, 1,8-cineole, camphor, β-citro-nellol, geraniol, linalool and α-citral that showed a fumigant effect against the *C. pipiens* adults [23,47], and also against other mosquitos. Zahran et al. (2011) [23] evaluated the adulticidal effect of twelve monoterpenes; they found that the monoterpenes tested against *C. pipiens* adults caused mortalities higher than 50% at the lowest concentration (10 mg/L) after 24 h of exposure. (R)-carvone and (R)-camphor showed high toxicity against adults at all three tested concentrations: they exhibited 73.3 ± 6.71% and 53.3 ± 8.88% of mortality at 10 mg/L, 86.7 ± 8.88% and 66.7 ± 8.88 at 50 mg/L air, 100 ± 0.0% and 90 ± 5.8% at 100 mg/L air after 24 h, respectively, while 80 ± 5.81% and 63.3 ± 3.36%, 96.7 ± 3.36% and 86.7 ± 8.88%, 100 ± 0.0% and 96.7 ± 3.36% of mortalities were recorded after 48 h, respectively. In previous literature, Rice and Coats (1994) [48] discovered that certain ketones such as carvone and pulegone were found to be more effective fumigants than alcohols. However, Zahran et al. (2011) [23] reported that menthol (monoterpene alcohol) had more adulticidal activity than camphor (monoterpene ketone) and camphene (monoterpene hydrocarbon) with mortality percentages of 63.3, 53.3 and 10%, respectively after 24 h of treatment. Against other insects, various EOs pure molecules were reported to be toxic fumigants. Rozman et al. (2007) [24] investigated the insecticidal effect of many monoterpenes such as 1.8-cineole, camphor, eugenol, linalool, carvacrol, thymol, borneol, bornyl acetate and linalyl acetate where 1,8-cineole was highly effective against adults of *Sitophilus oryzae* when it was applied at the lowest concentration (0.1 mL/720 mL volume) after 24 h of exposure, whereas, camphor was found to be more effective towards *Rhyzopertha dominica* with mortality of 100%. However, it has been documented that camphor presents a danger; it is a very toxic substance and many cases of camphor poisoning have been reported. Camphor poisoning could occur through inhalation, ingestion and also contact [49,50,51,52].

Otherwise, many other medicinal and aromatic herbs have proved their pesticidal potential against several insect families. In a recent review, Ebadollahi and his colleagues (2020) [53] highlighted the pesticidal efficacy of various plant-derived EOs obtained from the Lamiaceae plant family indicating *Agastache* Gronovius, *Hyptis* Jacquin, *Lavandula* L., *Lepechinia* Willdenow, *Mentha* L., *Melissa* L., *Ocimum* L., *Origanum* L., *Perilla* L., *Perovskia* Kar., *Phlomis* L., *Rosmarinus* L., *Salvia* L., *Satureja* L., *Teucrium* L., *Thymus* L., *Zataria* Boissier and *Zhumeria* Rech. In addition, Al-Harbi et al. (2021) [54] have evaluated the insecticidal effect of EOs isolated from *Ocimum basilicum*, *Nigella sativa* and *Lavandula angustifolia* on adults of *Sitophilus oryzae.* Against the *Culex* genus, Ramar et al. (2014) [55] investigated the insecticidal effect of twelve EOs including, for example, *Pimpinella anisum*, *Cinnamomum verum* J.S.Presl, *Cymbopogon nardus*, *Myrtus caryophyllus*, *Eucalyptus globulus* and *Pelargonium graveollens*.

All of these findings shed light on the promising insecticidal properties of EOs and their bioactive products. Until now, they are still the best alternative to synthetic chemical insecticides, especially with the emergence of resistant insect mosquitoes to different synthetic families [29,56,57]. Thus, plant-based natural products could be used as mosquitocidal botanicals in mosquito control programs, reducing the environmental and health toxic effect that the chemical products have created.

## 5. Conclusions

This study highlighted the insecticidal effect of *M. pulegium* and *R. officinalis* EOs, and their main compounds. The obtained results revealed the effectiveness variability between the tested substances, yet in general, they have all displayed fumigant toxicity. Based on the promising results, these natural products are recommended as suitable active products for possible botanical insecticides. However, further research should be conducted on the synergistic effect of these EOs and their bioactive compounds to enhance the mosquitocidal potential. The biosafety of these products on non-target organisms should be also evaluated.

## Figures and Tables

**Figure 1 plants-11-01193-f001:**
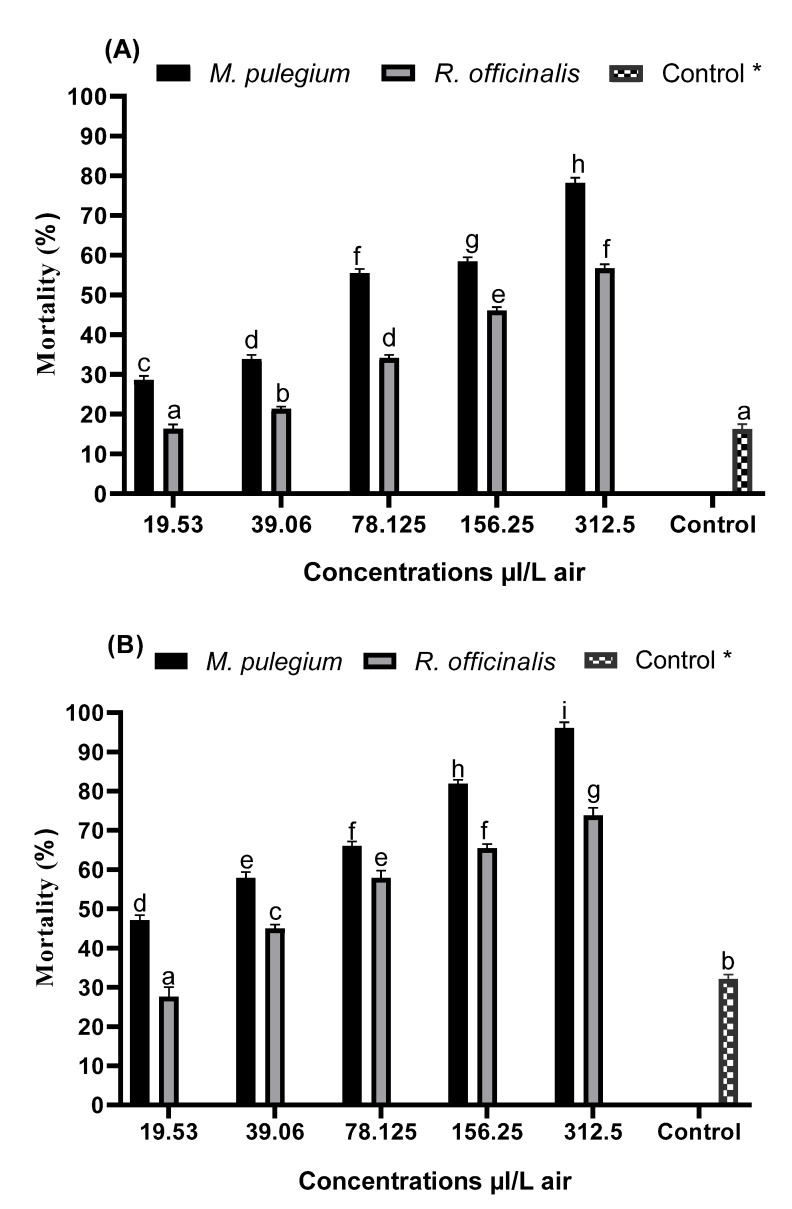
Evaluation of *M. pulegium* and *R. officinalis* EOs toxicity on *C. pipiens* adults according to different concentrations: (**A**): After 24 h; (**B**): After 48 h. (*): Positive control (Deltamethrin 0.05%). Means followed by the same letter are not significantly different at *p*-value (0.05).

**Figure 2 plants-11-01193-f002:**
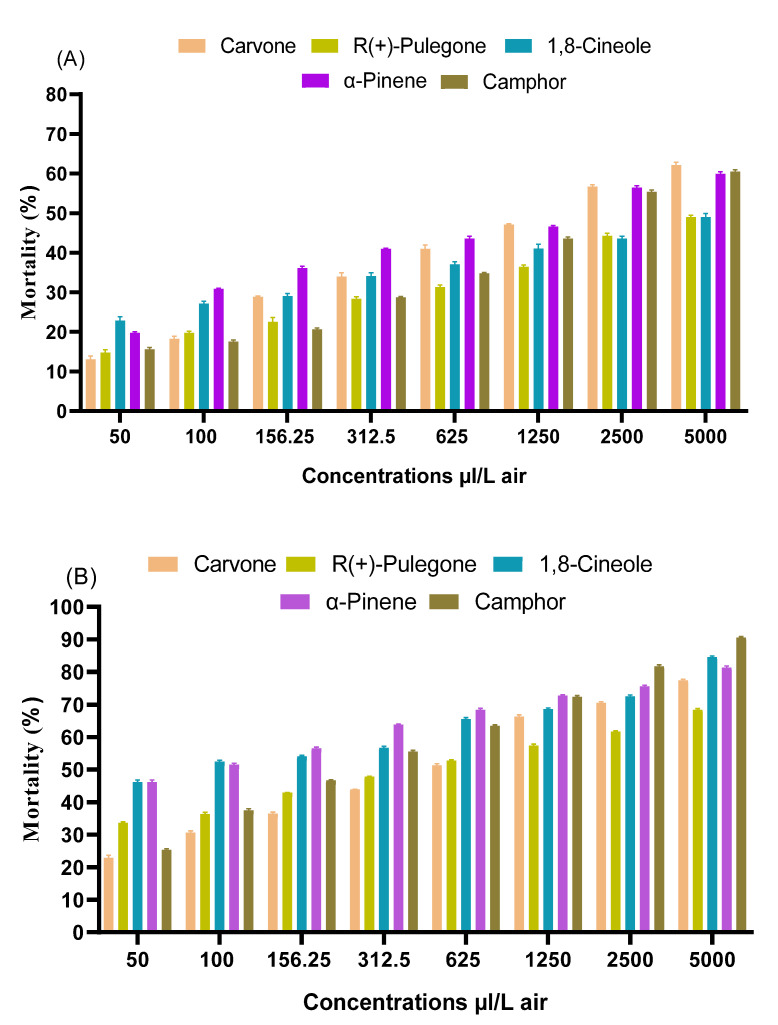
Toxicity of monoterpenes against *C. pipiens* adults at various concentrations after 24 h (**A**), and 48 h (**B**) of treatment. Results are means of three replicates (*n* = 3). Bars indicate standard errors. Tests were statistically significant at *p*-value (0.05).

**Table 1 plants-11-01193-t001:** The chemical compounds identified in M. pulegium and R. officinalis EOs.

**Compound**	**RI (Exp) ^a^**	RI (Lit) ^b^	** *M. pulegium* ** *****	** *R. officinalis* ** *****
α-Thujene	931	924	-	0.48
**α-Pinene**	939	932	0.42	**12.76**
Cyclohexanone-3-methyl	952	945	0.16	-
Camphene	953	946	-	2.47
β-Pinene	976	974	0.33	2.89
Unknown	957	-	-	t
Unknown	958	-	0.08	t
Unknown	961	-	0.07	t
Unknown	985	-	t	-
Myrcene	993	988	0.16	2.54
Octanol-3	993	995	1.32	-
Unknown	995	-	-	t
δ-2-Carene	1001	998	0.11	-
Unknown	1023	-	0.07	1.5
Limonene	1031	1029	1.34	-
**1,8-Cineole**	1033	1033	-	**29.31**
Unknown	1035	-	-	t
Unknown	1040	-	-	t
Unknown	1064	-	t	t
p-Mentha-3,8-diene	1071	1072	2.04	-
Unknown	1075	-	t	1.03
Unknown	1100	-	-	t
**Camphor**	1143	1141	-	**24.66**
Unknown	1148	-	-	t
Menthone	1154	1148	0.13	-
Borneol	1165	1166	-	5.46
Pinocarvone	1168	1160	1.21	-
Menthol	1173	1167	0.45	-
α-Terpineol	1185	1186	-	0.63
Dihydrocarvone	1194	1191	3.66	-
Myrtenol	1195	1194	-	1.23
Unknown	1207	-	0.09	-
**Pulegone**	1238	1233	**74.03**	-
**Carvone**	1242	1239	**5.45**	-
Unknown	1245	-	-	0.2
Peperitone	1252	1249	1.12	-
Unknown	1265	-	-	0.2
Myrtenyl acetate	1322	1324	-	0.22
α-Cubebene	1351	1345	-	0.15
α-Copaene	1381	1374	-	0.10
β-Bourbonene	1384	1387	-	0.16
Unknown	1397	-	0.06	-
Caryophyllene	1419	1417	0.35	-
γ-Gurjunene	1473	1475	-	0.2
Germacrene-D	1480	1484		0.42
Ledene	1493	1490	-	2.92
α-Muurolene	1499	1500	-	0.32
γ-Cadinene	1513	1513	-	0.8
Caryophyllene oxide	1581	1582	-	2.75
Copaen-4-α-ol	1584	1590	-	0.43
Tetradecanal	1611	1612	-	0,21
γ-Eudesmol	1630	1630	0.26	-
τ.Cadinol	1653	1654	-	0.19
α-Eudesmol	1649	1652	0.44	-
Total identified compounds			93.35	94.23

^a^: Experimental retention indices. ^b^: Retention indices from literature [30,34]. * The quantity of each compound was given by percentages that were presented as a ratio of the areas of the chromatographic peaks. T: trace < 0.05%. The compounds indicated in bold represent the major constituents, in both EOs, that have been tested for their insecticidal effect.

**Table 2 plants-11-01193-t002:** Lethal concentrations (LC_50_) of *M. pulegium* and *R. officinalis* EOs and their main compounds.

EOs and Monoterpenes	Exposure Time (h)	LC_50_ ^a^ (µL/L Air) (95% Confidence Intervals)	Slope ^b^	Intercept ^c^	R^2^	*p*-Value ^d^
* **M. pulegium** *	24	72.94 (60.34–83.32)	1.146 ± 0.15	2.865 ± 0.30	0.94	0.005
	48	25.43 (13.22–38.65)	1.169 ± 0.18	3.357 ± 1.23	0.94	0.005
* **R. officinalis** *	24	222.82 (210, 234.29)	0.957 ± 0.11	2.753 ± 0.63	0.93	0.003
	48	55.79 (50.77–60.81)	1.093 ± 0.08	3.091 ± 1.18	0.93	0.003
**Carvone**	24	1713.36 (1703.33–1723.36)	0.774 ± 0.04	2.497 ± 0.12	0.96	0.000
	48	538.96 (512.84–568.36)	0.827 ± 0.02	2.741 ± 0.07	0.99	0.000
**R(+)-Pulegone**	24	5395.58 (5295.58–5494.68)	0.515 ± 0.01	3.078 ± 0.02	0.99	0.000
	48	578.84 (578.24–597.64)	0.577 ± 0.05	3.406 ± 0.14	0.93	0.000
**1,8-Cineole**	24	5395.65 (5282.45–5486.55)	0.362 ± 0.01	3.649 ± 0.03	0.98	0.000
	48	84.96 (75.8–95.26)	0.537 ± 0.05	3.964 ± 0.14	0.90	0.000
**Camphor**	24	2269.64 (2190.41–2354.45)	0.663 ± 0.01	2.775 ± 0.04	0.98	0.000
	48	205.38 (167.83–300.51)	1. 027 ± 0.08	2. 6258 ± 0.22	0.94	0.000
**α-Pinene**	24	1294.64 (1230–1345.14)	0.642 ± 0.06	3.002 ± 0.16	0.92	0.000
	48	85.74 (74.04–98.21)	0.509 ± 0.01	4.016 ± 0.04	0.99	0.000

^a^: Lethal concentration killing 50% of exposed adult population. ^b^: Slope of the regression line ± SE. ^c^: Intercept of the regression line ± SE. ^d^: Significant effect at *p*-value < 0.05.

## Data Availability

All related data are within the manuscript.

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
