# Peer review of "Insecticidal Effect of Wild-Grown Mentha pulegium and Rosmarinus officinalis Essential Oils and Their Main Monoterpenes against Culex pipiens (Diptera: Culicidae)"

_plants, 2022, doi:10.3390/plants11091193_

Round 1

Reviewer 1 Report

The manuscript deals with a very important topic, plant-based insecticides.
The authors could give other examples of herbal drugs used as pesticides.
Do the authors have information on toxicity? can they irritate those who use it? This is important for information on the personnel who use these products.

Author Response

Dear reviewer,

The authors would like to thank the reviewer for the review of our manuscript entitled: “Insecticidal effect of wild-grown Mentha pulegium and Rosmarinus officinalis essential oils and their main monoterpenes against Culex pipiens (Diptera: Culicidae)”. We sincerely appreciate all valuable comments and suggestions.

Our responses to the Reviewers’ comments are described below. Appropriated changes, suggested by the Reviewers have been introduced to the manuscript (highlighted within the document with different color).

Comment 1:

The authors could give other examples of herbal drugs used as pesticides.

Response 1:

The authors provided other examples of herbs used as pesticides in the discussion part (Line 271-281 with yellow color).

“Otherwise, many other medicinal and aromatic herbs have proved their pesticidal potential against several insect families. In a recent review, Ebadollahi and his colleagues (2020) [54] have highlighted the pesticidal efficacy of various plant-derived EOs obtained from the Lamiaceae plant family indicating Agastache Gronovius, Hyptis Jacquin, Lavandula L., Lepechinia Willdenow, Mentha L., Melissa L., Ocimum L., Origanum L., Perilla L., Perovskia Kar., Phlomis L., Rosmarinus L., Salvia L., Satureja L., Teucrium L., Thymus L., Zataria Boissier, and Zhumeria Rech.  In addition, Al-Harbi et al (2021) [55] have evaluated the insecticidal effect of EOs isolated from Ocimum basilicum, Nigella sativa, and Lavandula angustifolia against adults of Sitophilus oryzae. Against the Culex genus, Ramar et al (2014) [56] investigated the insecticidal effect of twelve EOs including for example Pimpinella anisum, Cinnamomum verum J.S.Presl, Cymbopogon nardus, Myrtus caryophyllus, Eucalyptus globulus, and Pelargonium graveollens.”

Comment 2:

Do the authors have information on toxicity? Can they irritate those who use it? This is important for information on the personnel who use these products.

Response 2:

We thank the reviewer for this remark. Effectively, it’s very important to know the toxicity of these EOs to human health and the environment. The authors previously checked the information about the risk of these products. Indeed, Mint and Rosemary are among the herbs that are exempt from registration with the U.S. Environmental Protection Agency (EPA) under FIFRA section 25(b) regulations according to the report of Brian P. Baker and Jennifer A. Grant “Active Ingredients Eligible for Minimum Risk Pesticide Use: Overview of the Profiles”. This report listed the products that are so well established as safe and they don’t have to be registered. This helps the user to know that the product is made with ingredients that are safe and don’t present a danger to health. 

An extract from the report “Essential oils eligible for exemption as active ingredients in pesticides include cedarwood, citronella, cloves, geranium, mint, peppermint, rosemary, and thyme.”

Reviewer 2 Report

The paper entitled “Insecticidal effect of wild grown Mentha pulegium and Rosmarinus officinalis essential oils and their main monoterpenes against Culex pipiens (Diptera: Culicidae)”.  

The authors answered my question about camphor by arguing that it could be toxic to humans (+ references). This is very important information and should be given in a discussion.

Figure 2 should be enlarged as much as possible to make it more readable.

Line 257: menthol (alcohol). What kind of alcohol? For camphor, the authors give a monoterpene ketone in brackets. It is also monoterpene.

The text should be carefully checked for typographical errors.

Line 105 – what does this mean? C. pipiens (C. pipiens),

Doubled references in line 210 (Isman et al. 2011) [35].

Funding is not specified.

Reference 7 has only one author provided.

Author Response

Dear reviewer,

The authors would like to thank the reviewer for the review of our manuscript entitled: “Insecticidal effect of wild-grown Mentha pulegium and Rosmarinus officinalis essential oils and their main monoterpenes against Culex pipiens (Diptera: Culicidae)”. We sincerely appreciate all valuable comments and suggestions.

Our responses to the Reviewers’ comments are described below. Appropriated changes, suggested by the Reviewers have been introduced to the manuscript (highlighted within the document with different color).

Comment 1:

The authors answered my question about camphor by arguing that it could be toxic to humans (+ references). This is very important information and should be given in a discussion.

Response 1:

The authors indicated in the discussion part the toxicity of the camphor (highlighted within the manuscript with Green color). “Discussion part (Line 266-269)”

“However, it has been documented that camphor presents a danger; it’s a very toxic substance and many cases of camphor poisoning have been reported. Camphor poisoning could occur through inhalation, ingestion, and also contact [50, 51, 52, 53].”

Comment 2:

Figure 2 should be enlarged as much as possible to make it more readable.

Response 2:

Figure 2 has been enlarged.

Comment 3:

Line 257: menthol (alcohol). What kind of alcohol? For camphor, the authors give a monoterpene ketone in brackets. It is also monoterpene.

Response 3:

We thank the reviewer for this remark. Effectively, the menthol is a monoterpene alcohol. We indicated this in the text with green color (Line 258).

Comment 4:

The text should be carefully checked for typographical errors.

Response 4:

The authors carefully checked the manuscript for typographical errors.

Comment 5:

Line 105 – what does this mean? C. pipiens (C. pipiens),

Response 5:

Authors corrected the sentence; it’s actually Culex pipiens (C. pipiens).

Comment 6:

Doubled references in line 210 (Isman et al. 2011) [35].

Response 6:

Authors revised the manuscript and kept one reference in line 210.

Comment 7:

Funding is not specified.

Response 7:

Authors specified the funding.

Comment 8:

Reference 7 has only one author provided.

Response 8:

Authors corrected the reference 7.

Thank you again for your comments that helped us to improve our article quality. We hope that our responses will suit your expectations.

Sincerely.